# A Leucyl-tRNA Synthetase Urzyme: Authenticity of tRNA Synthetase Catalytic Activities and Promiscuous Phosphorylation of Leucyl-5′AMP

**DOI:** 10.3390/ijms23084229

**Published:** 2022-04-11

**Authors:** Jessica J. Hobson, Zhijie Li, Hao Hu, Charles W. Carter

**Affiliations:** Department of Biochemistry and Biophysics, University of North Carolina at Chapel Hill, Chapel Hill, NC 27599-7260, USA; jjean@email.unc.edu (J.J.H.); lizhijie53@gmail.com (Z.L.); enzyme@gmail.com (H.H.)

**Keywords:** genetic coding, protein synthesis, mechanistic enzymology, protein engineering, evolutionary intermediates, validating weak catalytic activities, single turnover kinetics, structural biology, evolutionary changes in the occupation of sequence space

## Abstract

Aminoacyl-tRNA synthetase (aaRS)/tRNA cognate pairs translate the genetic code by synthesizing specific aminoacyl-tRNAs that are assembled on messenger RNA by the ribosome. Deconstruction of the two distinct aaRS superfamilies (Classes) has provided conceptual and experimental models for their early evolution. Urzymes, containing ~120–130 amino acids excerpted from regions where genetic coding sequence complementarities have been identified, are key experimental models motivated by the proposal of a single bidirectional ancestral gene. Previous reports that Class I and Class II urzymes accelerate both amino acid activation and tRNA aminoacylation have not been extended to other synthetases. We describe a third urzyme (LeuAC) prepared from the Class IA *Pyrococcus horikoshii* leucyl-tRNA synthetase. We adduce multiple lines of evidence for the authenticity of its catalysis of both canonical reactions, amino acid activation and tRNA^Leu^ aminoacylation. Mutation of the three active-site lysine residues to alanine causes significant, but modest reduction in both amino acid activation and aminoacylation. LeuAC also catalyzes production of ADP, a non-canonical enzymatic function that has been overlooked since it first was described for several full-length aaRS in the 1970s. Structural data suggest that the LeuAC active site accommodates two ATP conformations that are prominent in water but rarely seen bound to proteins, accounting for successive, in situ phosphorylation of the bound leucyl-5′AMP phosphate, accounting for ADP production. This unusual ATP consumption regenerates the transition state for amino acid activation and suggests, in turn, that in the absence of the editing and anticodon-binding domains, LeuAC releases leu-5′AMP unusually slowly, relative to the two phosphorylation reactions.

## 1. Introduction

Our previous work built a new experimental framework for addressing the question of how peptides first became constructed according to what we now call the genetic code. We have argued that the early evolution of two superfamilies, or Classes, of protein enzymes called aminoacyl-tRNA synthetases (aaRS) [1], was closely coupled to that of the code itself [2,3,4,5] and to the progressive development of protein secondary and tertiary conformational characteristics [4,6]. We extend that work here by describing the biochemistry of a new addition to the set of evolutionary intermediates already described [7,8,9,10].

The Rodin and Ohno proposal that the two aminoacyl-tRNA synthetase Classes evolved from a single bidirectional gene [11] motivated us to resurrect ancestral aaRS by deconstructing genes of full length Class I and II enzymes and characterizing the catalytic properties of a hierarchical set of constructs containing their structurally conserved active sites [2,7,8,9,10,12,13]. Our previous work demonstrated activity for 46–(protozymes, [12]) and ~130–amino acid (urzymes, [7,8,9,10]) constructs excerpted from full-length tryptophanyl (TrpRS) and histidyl (HisRS)-tRNA synthetase proteins by preserving only those sequences consistent with bidirectional coding from opposite strands. Elsewhere in this volume [6], we present phylogenetic evidence that Class I aaRS have a mosaic structure consistent with genetically distinct and more ancient ancestry of the urzyme sequences and especially of the amino terminal protozyme segments containing their ATP binding sites.

The descent of contemporary enzymes from simpler ancestral forms that remain identifiable within extant structures [7,9,10,13] was not a radical suggestion. Indeed, from the continuous requirement for historical context during biological evolution—biological catalysts must have functioned continuously as they evolved from simpler to more sophisticated forms—it is by far the most conservative hypothesis. By consensus, the evident conservation of enzyme active sites across phylogenetic superfamilies implicates ancestral forms that strongly resembled contemporary forms.

However, experimental verification of progressive levels of catalytic proficiency has presented unprecedented difficulties:(i)Because we are ignorant of the stages by which ancestral active sites assimilated newer modules, we must use protein engineering to reconstruct ancestral forms.(ii)Urzymes are consequently inherently less stable than full-length counterparts because their construction eliminates substantial sources of stability.(iii)Exposure of extensive patches of hydrophobic residues complementary in structure to the deleted stabilizing mass renders urzymes less soluble, requiring solubilization as maltose-binding protein (MBP) fusions.(iv)Although transition-state stabilization is ~60% of that for full-length aaRS [7], urzyme catalysis is still 4–5 orders of magnitude weaker than full length aaRS, reducing experimental signal to noise.(v)Not knowing all requirements for activity contributes to poor reproducibility [14].

Similar difficulties challenge initial explorations of any previously uninhabited experimental landscape. Until recently, we alone have attempted to overcome them. However, Koji Tamura’s group at Tokyo University of Science [15] have now re-examined the catalysis we reported from protozymes encoded by opposite strands of a designed gene [12]. As we had described [12], in their hands the Class I protozyme catalyzed amino acid activation by ATP significantly more than maltose-binding protein (MBP) alone when incubated with various amino acids, whereas the Class II protozyme’s activity was closer to, but still greater than, that of MBP alone. That independent validation of our protozyme results [15] underscores the relevance of our work. Furthermore, both protozymes exhibited greater activity in the absence of any amino acid than did MBP alone. That promiscuous activity, together with results presented here, suggest that the aaRS protozymes and urzymes may catalyze other, non-canonical phosphoryl-transfer reactions.

Previously reported evidence for the authenticity of urzyme catalytic activity relied on three types of evidence: (i) significant burst sizes in the time dependence of ^32^P transfer from the γ-position of ATP into orthophosphate in the presence of pyrophosphatase [8,9,16] in single-turnover, active-site titrations (AST), (ii) pyrophosphate exchange assays that monitored ^32^P transfer from pyrophosphate into ATP in the presence of cognate amino acid, and (iii) mobility shift of [^32^P] labeled 3′ adenosine upon aminoacylation [7]. These assays were combined with mutation, genetic variation, and substrate concentration-dependence to attribute these activities to urzymes, and not to contamination.

We present here a parallel characterization of a third Class I urzyme based on *Pyrococcus horikoshii* leucyl-tRNA synthetase. For convenience, we use a nomenclature based on a four-module structure exhibited by all Class I aaRS (see the graphical abstract). The N-terminal segment containing the HIGH signature (the protozyme, ~46 residues) is denoted A, the variable-length insertion connecting peptide 1 (CP1) that follows the protozyme is B, the second two thirds of the Rossmann dinucleotide fold (specificity helix and KMSKS signature, ~84 residues) is C, and the C-terminal anticodon-binding domain is D. With this nomenclature, the LeuRS urzyme will be called LeuAC. It contains the most highly conserved modules composing the catalytic machinery, with rudimentary determinants for ATP [12] (A), amino acid [17], and tRNA substrates [18,19] (C). It lacks the B and D modules that enhance amino acid and tRNA substrate recognition.

LeuAC posed additional difficulties in compiling appropriate kinetic metrics:(i)Purified *P horikoshii* LeuRS and LeuAC both retain near stoichiometric amounts of bound leucyl-5′AMP [20], complicating straightforward interpretation of the [leucine]-dependence of pyrophosphate exchange assays.(ii)LeuAC and MBP cannot be separated cleanly following TEV cleavage because the urzyme and MBP cleavage products aggregate.(iii)It is difficult to prepare cognate tRNA^Leu^ with >30% acylatability [20,21,22,23,24], increasing the uncertainty of kinetic parameters for aminoacylation.(iv)Use of [α^32^P] ATP to monitor all three nucleotides—ATP, ADP, and AMP—during amino acid activation by both LeuRS and LeuAC reveals significant production of the non-canonical nucleotide [^32^P] ADP, posing unexpected mechanistic puzzles.

It is crucially important that TEV cleavage significantly enhances catalysis of amino acid activation and tRNA^Leu^ aminoacylation by LeuAC, that an AMSASA mutant of the catalytic KMSKSK signature alters both activation and acylation rates, and that saturating substrate concentrations differentiate LeuAC from LeuRS. LeuAC therefore fulfills these criteria for attributing relevant catalytic activities: (i) significant burst sizes in single turnover experiments for both half reactions, (ii) pyrophosphate exchange activity, (iii) aminoacylation of tRNA^Leu^, (iv) binding significant amounts of ^14^C leucine—presumably as leucyl-5′AMP, and (v) modification (TEV cleavage) of the urzyme construct itself materially enhances observed activities, whereas active site mutation reduces it. These results unequivocally authenticate LeuAC catalytic activity.

Finally, we describe the catalytic production of ADP, a noncanonical product first observed in a rarely cited study of phosphoryl-transfer reactions exhibited by ArgRS, ValRS, PheRS, and AspRS [25]. Mechanistic studies in the intervening years make it likely that the original interpretation of ADP formation in forming enzyme-bound aminoacyl-thioester is not correct. A likely alternative is that ATP phosphorylates the 5′-leucyl AMP intermediate in situ, rebuilding the PPi leaving group into a transition-state-like ligand.

For these reasons, the variegated and unusually interdependent experiments described here furnish an essential to guide further work in this challenging area.

## 2. Results

### 2.1. LeuAC Is a Novel Class I Urzyme Derived from a Large Class IA aaRS

The LeuAC urzyme, derived from the *P. horikoshii* LeuRS, had not been fully characterized, and its provenance in our laboratory [26,27] was distinct from those of the TrpRS and HisRS urzymes. The original Rosetta designs (Ozgun Erdogan, unpublished data) were performed according to a two-way, three-factor factorial design to test the relative importance of (i–ii) the linker lengths where the two inserts, connecting peptides 1 and 2 (CP1 and CP2) were removed, and (iii) to assess the usefulness of a Rosetta module designed to minimize development of surface hydrophobic clusters that made the TrpRS urzyme hard to solubilize [28]. Eight LeuAC constructs were purified and assayed for PPi-exchange activity and solubility. Here, we characterize the construct with the greatest PPi-exchange activity, for which we also had assayed amino acid specificity [26,27]. CP1 and CP2 deletions replaced by a single peptide bond had slightly better functionality than those with additional amino acids, and the Rosetta solubility score was actually inversely correlated with observed solubility (M. Collier unpublished).

#### 2.1.1. The MBP-LeuAC Fusion Is Pure Enough to Exclude Contaminating Activities

TEV cleavage of the MBP-LeuAC fusion protein is essentially complete (Figure 1a). Estimates derived from band intensities (Figure 1b) using tryptophan residues in each relevant species (MBP-LeuAC fusion, 10; LeuAC, 3; MBP 7; *E. coli* LeuRS, 24) suggest that TEV cleavage leaves only 6% of the fusion protein intact. The absence of any band with the molecular weight of *E. coli* LeuRS (Figure 1b) implies that the maximum amount of full-length LeuRS would be, at most, 1 part in 10^4^. Estimates of the active fraction of LeuAC molecules (0.4–0.6) in Figure 1c are more than 4000 fold greater than this value. Moreover, the contribution of maltose-binding protein to the amino acid activation reaction has been measured at less than 10^−5^ times values reported here [12,15]. Thus, neither full-length LeuRS contamination nor MBP can account for any catalytic activities documented here.

#### 2.1.2. TEV-Cleaved LeuAC Fusion Protein Remains Non-Covalently Bound to MBP

We enriched the LeuAC by first purifying on amylose, cleaving the MBP fusion, then binding the his-tagged LeuAC to nickel. The resulting LeuAC, however, was not entirely free of MBP, and precipitated when concentrated. Concentrations high enough for single-turnover assays or NMR studies could not be achieved. For this reason, all experiments described subsequently are for either the MBP LeuAC fusion protein itself or the TEV-cleaved, unseparated protein.

### 2.2. Both LeuRS and LeuAC Retain Tightly Bound Leucyl-5′ AMP

Amino acid-dependence of pyrophosphate (PPi) exchange is an important criterion for confidence in the authenticity of catalytic activity. Lack of that dependence for LeuAC PPi exchange activities created concern. Previous studies [20] showed that full-length LeuRS purifies with near stoichiometric amounts of bound leu-5′AMP. Those authors incubated full-length *E. coli* LeuRS with excess tRNA^Leu^ to obtain sufficient unliganded LeuRS for studies of the partitioning of pre- and post-transfer editing. That approach was impractical owing to the relatively low stability of LeuAC in 2M urea necessary to dissociate tRNA^Leu^ from the acylation complex.

Extended dialysis of MBP-LeuAC fusion protein against six successive 1000-fold volumes of 40 mM Tris-HCl buffer, pH = 8.0, 100 mM NaCl, 20% glycerol, 10 mM BME increased PPi exchange 1.7 fold (*p* = 0.01). Equilibrium dialysis is inefficient at removing tightly bound ligands because the equilibrium concentration of free ligand inside the membrane is so low that the gradient across the membrane fails to remove much of the bound ligand, even after extensive volume changes. The dialysis experiment nonetheless implicates the presence of a tightly bound ligand. Nitrocellulose binding and gel filtration (Section 2.3.3) suggest that this ligand is leucyl-5′ AMP, which can support [leucine]-independent PPi exchange. Persistence of that ligand through extensive dialysis is consistent with the expected ~1000-fold increase in affinity of the 5′adenylate, relative to the free amino acid, with pre-steady-state bursts observed in single-turnover assays (Figure 2 and Figure 3), and with lower limits of affinity from size exclusion chromatography.

### 2.3. LeuAC Accelerates Amino Acid Activation

Aminoacyl-tRNA synthetases establish covalent bonds between each amino acid and the set of anticodons necessary to read messenger RNAs according to the genetic coding table. That “assignment catalysis” combines multiple molecular recognition (amino acid and tRNA) and catalytic steps. Two chemical steps must be accelerated to make these assignments. The amino acid carboxyl group must be activated by ATP, forming an aminoacyl-5′AMP and the activated amino acid must be transferred to the correct (i.e., “cognate”) transfer RNA. AaRS urzymes excerpt from the full-length enzymes what seem to be the minimal catalytic apparatus necessary to accelerate both reactions.

Removing substantial amounts of the full length enzymes reduces their functionality two ways: (i) acceleration of activation and acyl-transfer are ~10^−5^ and 10^−3−^fold slower, respectively, relative to full-length counterparts [7,13], and (ii) their substrate specificities are substantially relaxed [26,27]. Detailed characterization of urzymes ultimately requires validating both their catalytic proficiencies and their specificities. We focus here on the former. The catalytic power of the full-length enzymes is so extraordinary that even with these losses, urzymes retain rate accelerations of ~10^9^ and 10^6^-fold.

We inferred the authenticity of catalysis by previously described urzymes [8,9] from (i) active-site titration assays showing that a significant fraction of the protein molecules contribute to the accelerated reactions [16], (ii) significantly different K_M_ values from Michaelis–Menten kinetics experiments, and (iii) altered catalysis induced by mutation. Each kind of evidence uses a different logic to implicate the main component in complex mixtures that might contain small amounts of much more active wild-type enzyme or other contaminating catalysts. (i) Shows that the fraction of molecules contributing to catalysis is comparable to the main protein component (Section 2.3.1 and Section 2.4); (ii) rules out the wild-type enzyme as the catalyst, which would saturate at substrate concentrations equal to the wild-type K_M_ (Section 2.5); and (iii) modifies uniquely the putative catalyst, so altered activity independently confirms authenticity (Section 2.3.2 and Section 2.6).

The LeuAC urzyme satisfies each of these criteria as an authentic catalyst of both steps in the synthesis of leu-tRNA^Leu^. Experimental replication has afforded more detailed statistical significance of the resulting evidence than we reported previously, thereby strengthening our previous conclusions [7,13].

#### 2.3.1. Single Turnover Kinetic Experiments Establish the Stoichiometry of Catalysts

The urzymes we have studied all bind tightly to the aminoacyl-5′AMP intermediate product, so that product release is rate-limiting in single-turnover enzymatic assays with substrate-level amounts of enzyme [8,9]. Under these conditions, the burst size, or amplitude of the exponential decay of substrate, can be used to estimate how many catalytically active molecules the added enzyme contains [16,29]. [γ^32^P] ATP is rapidly lost at different rates over a ~90 min time course in the presence of full-length LeuRS, MBP-LeuAC fusion protein, or TEV-cleaved LeuAC. With [α^32^P] ATP LeuRS produced not only the expected [^32^P] AMP, but also [^32^P] ADP (Figure 2a), which LeuAC produced in near stoichiometric amounts (Figure 2b).

Routine use of AST assays—to characterize preparations of MBP-LeuAC fusion proteins, and in searching for high-affinity inhibitors—produced highly replicated data (Table 1), from which we can compare behaviors of the fusion protein and its TEV-cleaved products, MBP and LeuAC. TEV-cleaved LeuAC produces higher first-order rates and larger burst sizes for the loss of ATP and the appearance of the two other adenine nucleotides than does the fusion protein (Figure 3c). Replicated AST assays with [α^32^P] ATP (Table 1) verified the high statistical significance of these observations (Figure 3). Two predictors, TEV-cleavage and AMP, explain 93% of the variance in activation free energies for the first-order rates, ΔG^‡^k_chem_, with *t*-test P values of ~2 × 10^−10^ and 2 × 10^−7^.

#### 2.3.2. TEV-Cleavage of the MBP Fusion Protein Enhances the First-Order Rate of ATP Consumption

TEV cleavage reduces ΔG^‡^k_chem_ for all three reactions by ~0.5 kcal/mole, and first-order rate of AMP production is further increased by a similar amount from that observed for the loss of ATP and appearance of ADP. Consequently, the initial appearance of AMP occurs with a rate slightly greater than the loss of ATP or the appearance of ADP (Figure 3a). The dependence of the amplitude parameter, A, representing the pre-steady-state burst size, is also significantly suppressed in the MBP fusion protein (Figure 3b) relative to that of TEV-cleaved LeuAC. The larger burst sizes, together with the significant functional modification in the fusion protein, strongly and independently imply that the time courses in Figure 2c cannot be attributed to contaminating catalysts, and therefore arise from LeuAC itself.

Timecourses for AMP production differ in detail between full-length LeuRS and TEV-cleaved LeuAC (Appendix A). The amplitude parameter is markedly reduced for TEV-cleaved LeuAC (Appendix A). Steady-state turnover behaviors also differ (Appendix A). TEV-cleaved LeuAC exhibits increased turnover after the ADP production plateaus. This curious and reproducible observation may be related to mechanistic issues discussed in Section 3.3.

Near stoichiometric appearance of the non-canonical product, ADP, is unexpected considering the consensus mechanisms of amino acid activation by aaRS, which proceeds by the concomitant release and hydrolysis of PPi by the pyrophosphatase present in the assay mix, but which cannot be traced if the [^32^P] label is in the γ position. It recalls an early, generally overlooked, publication from Zamecnik’s laboratory [25] that some aaRSs also catalyze ADP production. ADP production by the full-length enzyme represents a small fraction (~3%) of the total active-site concentration (Figure 2a). The protein mass missing in LeuAC reduces the production of ADP, so LeuAC produces near-stoichiometric ADP. The high amplitude of this unexpected, non-canonical product helps to validate the authenticity of LeuAC catalytic functionality, but poses significant mechanistic questions, which are discussed further in Section 3.3.

#### 2.3.3. Mutual Perturbations by LeuAC and Substrates in Size Exclusion Chromatography Implicate a Significant Fraction of Active Protein

ADP production compromises the straightforward interpretation of active-site titrations in terms of the number of active sites. The resulting ambiguity underscored the need for an orthogonal measurement of the active fraction of enzyme in preparations of both full-length *P. horikoshii* LeuRS and LeuAC derived from it. For this purpose, we first measured the retention of [^14^C] leucine bound to LeuRS and LeuAC collected on nitrocellulose filters [16]. Although LeuRS retained 0.48 ± 0.13 moles of [^14^C] leucine/mole enzyme, LeuAC was much less reproducible, retaining 0.10 ± 0.07 moles/mole enzyme. For that reason, we also used size exclusion chromatography on Sephadex G15 to separate TEV-cleaved LeuAC from low molecular weight substrates. G15 Sephadex was chosen because both LeuAC and ATP remain, anomalously, in the included volume in G25 whereas both LeuAC and ATP are excluded from Sephadex G10.

G15 Sephadex size-exclusion profiles of LeuAC, ATP, and reaction mixtures with and without 5 mM leucine were recorded at 280 nm (Figure 4a) and that of ATP was also recorded at A260. Although the stoichiometric excess of ATP in the reaction mixtures contributed significantly to the absorbance profile, the molar extinction coefficients of ATP (2.21 × 10^3^) and LeuAC+MBP (9.63 × 10^4^) are in the ratio = 0.024. Thus, ATP bound to LeuAC is virtually undetectable at 280 nm, allowing quantitation of both eluted products. Integrated A_280_ of the two reaction mixtures (26.1 and 28.2) are both within experimental error of the total A_280_ (27.6) of the LeuAC (7.8) and ATP (19.8).

Total CPM from [^14^C] leucine and A280 absorbance profiles for both reactions were scaled together to evaluate difference profiles (Figure 4b). Quantitative deconvolution of peak integrals, described in Methods, demonstrate that LeuAC and leucine reciprocally shift each other’s elution toward the excluded volume. Addition of 5 mM leucine to the reaction mixture shifts ~4.9 μM (58%) of the 8.5 μM LeuAC to the left (green profile); whereas addition of 8.5 μM LeuAC shifts ~7.3 μM (14%) of the 50 μM or of the [^14^C] leucine to the left (blue profile with histogram). Leucine and LeuAC perturb comparable proportions of the other species. Nearly 60% of the LeuAC binds reversibly to exogenous leucine.

Inasmuch as the [^14^C] leucine-induced shift in elution volume is about 40% of the difference between the eluted positions of LeuAC and ATP, the half-life of bound leucine (~21 min) is likely a comparable fraction of its transit through the column. Quantitative analysis of this behavior would have required that the elution buffer contain constant ligand concentrations [31] as described by Hummel and Dreyer, which can reveal tight binding interactions [32], but was impractical here owing to the cost of radiolabeled ligand [33]. Qualitative analysis sets a lower limit on both the stoichiometry and affinity of the active LeuAC•leucyl-5′ AMP complex (Figure 4c). Qualitatively, the ligand off-rate must be slow enough to account for displacement of [^14^C] leucine equivalent to 85% of the LeuAC concentration by ~ half the difference between its unperturbed elution and the protein peak (Figure 4b), suggesting a half-life of up to 10 min for the LeuAC•leu-5′ AMP complex. For any reasonable on-rate, that half-life sets an upper limit on K_D_ (i.e., <10^−7^ M; dashed line in Figure 4c) that is two orders of magnitude tighter than the measured K_M_ value in Figure 9b. This estimate sets a lower bound on the affinity of the bound form of [^14^C] leucine because detection required using concentrations substantially higher than the expected dissociation constant (<10^–8^ M). Reciprocal perturbation of SEC profiles by LeuAC and its leucine and ATP substrates imply catalysis of leucine activation. Figure 4b is therefore consistent with a bound adenylate.

#### 2.3.4. The LeuAC MBP Fusion Protein Catalyzes Pyrophosphate Exchange

We compared PP_i_ exchange activity of MBP-LeuAC and full-length LeuRS in the two-level, three-factor experiment in Table 2. Regression analysis in Figure 5 shows that β coefficients for the regression model are all statistically significant, with *t*-test values < 0.01. Catalysis by LeuRS is, on average, ~1500 times faster than that by LeuAC, owing to the 4.3 kcal/mole difference between mean activation free energies (Figure 5a). Qualitative differences between the LeuRS and LeuAC activities are indicated by the two, 2-way interaction free energies. Potassium fluoride (KF) is used in PPi exchange assays to eliminate activity by contaminating phosphatases (18). LeuAC-catalyzed incorporation of [^32^P] PPi into ATP is inhibited by KF, whereas that by LeuRS is unaffected (Figure 5b). LeuRS is stimulated by excess leucine, whereas LeuAC is not (Figure 5c). Separate tests for enzymatic hydrolysis of [^32^P]-labeled ATP and PPi by the MBP-LeuAC fusion at the concentrations used in these experiments (3 μM) did not exceed the background (Appendix A). Nonetheless, the experiment summarized in Figure 5 does not, by itself, definitively authenticate the biologically relevant PP_i_ exchange activity, because it does not demonstrate leucine concentration dependence.

### 2.4. LeuRS, LeuAC MBP Fusion, and TEV-Cleaved LeuAC Catalyze tRNA^Leu^ Aminoacylation

From many perspectives, the key aaRS urzyme catalytic activity is acyl transfer of activated amino acid to cognate tRNA. Here, we demonstrate catalysis of acyl-transfer by LeuAC for the first time (Figure 6a). TEV cleavage increases LeuAC aminoacylation rates by 10-fold (Figure 6b). The biphasic time dependence of aminoacylation is described in the following paragraph. An ensemble of 28 acylation assays (Appendix A) reveals that the principal determinants of ΔG^‡^(k_chem_) are (i) whether the catalyst is full length LeuRS or LeuAC urzyme and (ii) whether or not the urzyme is the MBP fusion or is TEV-cleaved (Figure 6c). TEV cleavage increases initial rates of aminoacylation ~10 fold. Β-coefficients for the overall effect of TEV cleavage (–1.36 kcal/mole; Figure 6b) and its effect on ΔG^‡^k_chem_ (–1.16; Figure 6c) suggest that about 85% of the increase derives from enhancement by TEV cleavage of the single-turnover rate, k_chem_, rather than from increases in turnover, k_cat_.

Time courses for aminoacylation by both LeuRS and LeuAC (Figure 7a) both exhibit bi-phasic kinetics that fit with very small unexplained variances (R^2^ > 0.98) to Equation (1) for a first-order decay of a single-turnover and steady state turnover, (Figure 7b). The fitting precision; the physical interpretation of β coefficients as the amplitude of the first-order phase, A, and the first-order and steady-state rates, k_chem_ and k_cat_, and the roughly parallel changes in the three parameters in amino acid activation and acyl-transfer (Appendix A and Figure 2, Figure 3, Figure 6 and Figure 7) argue that they are appropriate metrics. Comparison of these metrics for LeuRS, MBP LeuAC fusion, and TEV-cleaved LeuAC are, in turn, evidence for the authenticity of amino acid activation and tRNA acylation by all three catalysts.

### 2.5. LeuAC Michaelis–Menten Kinetic Parameters Distinguish Its Activity from That of LeuRS

Figure 8 shows experimental Michaelis–Menten data comparing the steady-state dependence of aminoacylation by *P. horikoshii* LeuRS and TEV-cleaved LeuAC on tRNA^Leu^ and leucine. Despite the obstacles noted elsewhere to steady-state kinetic analysis, these data help to confirm the distinct behavior and thus the authenticity of LeuAC catalysis. In particular, saturation of LeuRS and LeuAC by [tRNA^Leu^], which were performed using the same tRNA sample, differ by an order of magnitude in K_M_. Saturation of LeuAC could not be achieved, owing to the high K_M_ and the low total acylatability (0.28) of the tRNA substrate (Figure 8a). Discrepancies between K_M_ values published by the same group for *P. horikoshii* LeuRS [34,35] led us to re-determine that value (Figure 8a).

The Michaelis–Menten plot for [tRNA^Leu^]-dependence is only marginally better than a linear fit. Curvature can nonetheless be demonstrated by the quadratic dependence of residuals of the Michaelis–Menten model compared with those for a linear fit (Appendix A). K_M_ values for leucine-dependence (Figure 8b), though similar (9.5 ± 1.3 × 10^−6^; LeuAC vs. 5.9 ± 0.8 × 10^−6^ for LeuRS), also differ by a statistically significant 2.8σ.

### 2.6. Product Release Is Rate-Limiting for Both Amino ACID activation and tRNA^leu^ Aminoacylation by LeuAC

Active-site mutations are a definitive test for the authenticity of catalysis by urzymes because significantly altered activity changes necessarily imply that the urzyme construct is the source of observed catalytic activities. Accordingly, we prepared a mutant LeuAC in which both lysine residues of the KMSKS signature were mutated to alanine. A third lysine following the final serine of the signature motif was also mutated to alanine as a precaution against the possibility that the LeuAC KMSKS loop is flexible enough to employ its charge in transition-state stabilization. Reductions in catalytic activity are statistically very significant (*p* < 0.003) (Table 3). Notably, the activation free energy, ΔG^‡^k_chem_, for activation also depends strongly on TEV-cleavage of the fusion protein. That dependence was not tested for aminoacylation. The AMSASA mutant behavior reinforces the authenticity of both reactions.

Remarkably, the wild-type lysine residues contribute only modestly to either rate acceleration. Appendix A have the design matrices from which the regressions summarized in Table 3 were derived. Data in those tables show that activation and aminoacylation by WT LeuAC are only ~2.5× and ~30% faster than the AMSAS mutant. Implications of this important observation are developed in Section 3.4.

### 2.7. Product Release Is Rate-Limiting for Both Amino Acid Activation and tRNA^leu^ Aminoacylation by LeuAC

Single turnover kinetics furnish three metrics by which LeuRS, LeuAC fusion protein, and TEV-cleaved LeuAC all differ. Despite the ambiguity induced by ADP production, the amplitude parameters furnish qualitative, but decisive evidence that the observed catalysis arises from the major component within each sample.

Curiously, k_chem_ values for aminoacylation by full-length LeuRS and TEV-cleaved LeuAC both are faster than that of the MBP fusion protein (Figure 6c). That pattern—positive β-coefficients for the urzyme, relative to full-length and negative β-coefficients for TEV cleavage—recurs in several contexts; it provides important and orthogonal evidence for the authenticity of LeuAC catalysis. As with amino acid activation, Figure 6c shows that the MBP-LeuAC fusion protein is the weakest catalyst (β_Urz_ = +1.1 kcal/mole), TEV cleavage more than compensates for this (β_TEV_ = –1.2 kcal/mole).

Finally, catalytic time courses furnish a unique estimate for the ratio, (k_chem_/k_cat_), of rates for unimolecular conversion of enzyme•substrate to enzyme•product, and product release. In contrast to the [^32^P] ATP consumption experiments summarized in Figure 2, Figure 3 and Figure 4, which were intentionally active-site titrations and had a 10:3 ratio of ATP to catalysis, acylation experiments listed in Appendix A, were done at a wide range of different tRNA^Leu^ and enzyme concentrations. For that reason, distributions of the amplitude parameter, A, in Figure 1c and ΔG^‡^(k_chem_/k_cat_) in Table 3 include only experiments in which the [tRNA^Leu^]/[Enzyme] ratios were in the range 0.5–5.0 to approximate single turnover conditions.

Table 4 identifies differences between ΔG^‡^(k_chem_/k_cat_) for LeuRS, LeuAC-MBP fusion, and TEV-cleaved LeuAC. Parameters for all three catalysts change consistently between amino acid activation and tRNA^Leu^ aminoacylation. LeuRS and TEV-cleaved LeuAC both have k_chem_ values ~55-fold greater than k_cat_, consistent with the appearance of a pre-steady state burst in both reactions. Curiously, although values for full length LeuRS are larger, the ranges of values observed imply that the differences are statistically significant only for the activation reaction.

## 3. Discussion

### 3.1. LeuAC Is an Authentic Catalyst of Both Amino Acid Activation and tRNA Acylation

Neither the amplitude values, A, for [^32^P] ATP consumption nor those for tRNA aminoacylation can be converted into active-site titers, for different reasons. The n-values derived from loss of [^32^P] ATP are corrupted by the conversion of ATP into ADP (Figure 2b,c). Burst sizes from single-turnover acylation are uncertain because the low concentration of unacylated tRNA^Leu^ means that an unknown fraction of inactive tRNA may be inhibitory. The burst sizes for both amino acid activation and tRNA^Leu^ aminoacylation are nonetheless about four orders of magnitude greater than can be attributed to contaminating activities.

Moreover, ATP consumption (Figure 3), by which we previously estimated the active-site titer, pyrophosphate exchange (Figure 5), and tRNA^Leu^ aminoacylation by LeuAC (Figure 6) are both significantly enhanced by TEV cleavage. Enhancement of ATP-dependent leucine activation is about two-fold, compared with an uncertainty of ~0.06, leading to highly significant *p*-values (10^-10^) under the null hypothesis. Cleaved LeuAC increases aminoacylation of tRNA^Leu^ by an order of magnitude (Figure 6b,c), although the statistical significance (*p* = 0.003) is lower. The ratio of first-order to steady-state rates, given by ΔG(k_chem_/k_cat_), are similarly enhanced by TEV cleavage (Table 3). These enhancements cannot be attributed to treating contaminating cellular enzymes with TEV protease. Finally, although we have not assayed it as extensively as we have the original LeuAC1, the LeuAC2 construct has mutations in 19 sites that increase solubility without detectable changes in active-site titration (P~0.99; Table 1).

### 3.2. Near Quantitative Production of ADP Poses Novel Mechanistic Questions

High ADP production is another novel result from this work. A high ratio of ATP consumed per activated aminoacyl-5′AMP produced is generally observed in the presence of an incorrect amino acid, and is diagnostic for hydrolytic error correction [36]. That appears not to explain the time courses in Figure 2. Early work by Zamecnik’s laboratory documented the catalytic production of noncanonical adenine nucleotides by several aaRS including Class I ValRS, ArgRS and Class II AspRS and PheRS [25]. Zamecnik’s group suggested phosphorylation of the aminoacyl-5′AMP to account for the appearance of ADP in their experiments, which they interpreted in terms of reacting with an enzyme-bound thioacyl intermediate. None of the structural and/or mechanistic studies of aaRS in the intervening years have implicated transient formation of aminoacyl-thioenzyme intermediates during aminoacylation.

The source of the labeled ADP product in this work is [α^32^P] ATP, rather than either [^3^H] or [^32^P] labeled AMP and/or ADP. Full-length LeuRS from both *E. coli* and *P. horikoshii* produce minor amounts of ADP. However, LeuAC produces stoichiometric amounts of ADP along a time course closely mirroring that for ATP consumption (Figure 2c). ADP production thus seems most likely to result from phosphoryl transfer from ATP to an enzyme bound group. The eventual production of both [α^32^P] AMP with α-labeled ATP and [^32^P] inorganic phosphate when the phosphoryl donor is [γ^32^P] ATP suggest in situ phosphorylation of aminoacyl 5′-AMP. The cycle of reactions in Figure 9a reconciles the unexpected data in Figure 3 with the substantial evidence adduced here for LeuAC catalysis of the canonical reactions—leucine activation and tRNA^Leu^ aminoacylation—associated with full-length aaRS. The essence of this mechanism is that two molecules of ATP are expended to regenerate what is actually the transition-state configuration for leucine activation (salmon-colored rectangle, Figure 9a) before the [α^32^P] label appears as AMP.

We propose that the specificity for the LeuAC tRNA^Leu^ 3′-terminal adenosine binding pocket is relaxed sufficiently to accommodate ATP—perhaps in the site normally occupied by the tRNA^Leu^ 3′-terminal A76—in a manner conducive to phosphorylation of the bound adenylate (thick black arrows). This mechanism is consistent with the puzzling result that *n*-values from the amplitudes of active-site titration [γ^32^P] ATP are close to 2.0 and with near simultaneous, stoichiometric consumption of ATP with production of ADP (Figure 3c). However, it seems at first glance to involve unlikely chemistry. (i) The Pα in the Leu-5′AMP mixed anhydride itself does not furnish a convincing nucleophile. (ii) The intermediate phosphorylation product, Leu-5′ADP—a double anhydride—would be quite high energy, hence labile. (iii) Both the double anhydride and the successive pileup of phosphoryl groups appear to lack chemical precedent.

Neither the double anhydride nor the successive phosphorylations are entirely without precedent, however. One case in point is carbamoyl phosphate synthetase which synchronizes the utilization of two ATP molecules in the synthesis of the high energy carbon and ammonia donor, carbamoyl phosphate [37]. In that reaction, both γ phosphates are transferred, in succession (but at different sites) to similar moieties. The first phosphorylates bicarbonate, the second carbamate, producing two molecules of ADP per round. A similarly puzzling example are the Coenzyme A transferases [38], in which two-step acyl exchange reactions involving bound thiocarboxylate esters proceed via enzyme-bound high energy anhydride intermediates. Citrate is converted to oxaloacetate and acetate, via an acetyl-citryl anhydride intermediate [39,40], whose free energy has been estimated at ~–22 kcal/mole [41], which is comparable to that expected for Leu-5′ADP and Leu-5′ATP.

### 3.3. Promiscuous ATP Binding Sites May Enable In Situ Phosphorylation of Leu-5′AMP

Structural biology supports the plausibility of the dual phosphorylation steps (Figure 9b). The tRNA^Leu^ complex of full-length *P. horikoshii* LeuRS (1WZ2) furnishes a basis for positioning the 3′-terminal adenosine, A76, within the active site of both full length LeuRS and LeuAC. Atomic coordinates of that adenosine allowed us to explore different ATP conformations by superimposing the respective adenine ring systems. A variety of ATP conformations have been described in the literature, including molecular dynamics (MD) simulations of either Mg^+2^•ATP [42] or ATP alone [43] in water. Both reports identify extended ATP conformations, similar to those observed in protein active sites, in which the glycosyl (χ) and ribose-phosphate (γ) bond torsion angles fall within a narrow range [(χ, γ) = (–150, 60)]. Simulations in the absence of Mg^+2^ ion in water, however [43], identify two additional stable conformations [(χ, γ) = (60, –60); ATP-1], and [(χ, γ) = (75, –150); ATP-2]. Using Chimera [44] to adjust ATP-1 and ATP-2 to the appropriate (χ, γ) angles, we constructed all three conformations and superimposed their adenine rings onto that of A76. To our surprise, the ATP-2 and ATP-1 configurations placed the corresponding γ-phosphates into open pockets in the active site, such that they are almost ideally positioned to phosphorylate first the α-phosphate and then the (resulting) β–phosphate (Figure 9b) as shown in Figure 9a.

Given that both LeuRS and LeuAC retain substantial amounts of leucyl-5′ AMP, that ligand is likely the initial state of both catalysts in all assays. The amplitude of AMP formation (~5% of total LeuAC; Figure 2c) suggests that the off-rate for the leucyl-5′AMP is sufficient to enable it to dissociate, leading rapidly to hydrolysis and rebinding of [^14^C] leucine under the saturating concentrations of ATP and leucine, and to incorporation of the label as [α^32^P] AMP into the adenylate (thick green arrows), in keeping with our interpretation of Figure 4b.

If the in situ phosphorylation represented in Figure 9a was required for PPi exchange by LeuAC, that would make sense of the two-way interactions in Figure 5b,c. Inhibition by KF of [^32^P] ATP synthesis from [^32^P] PP_i_ by LeuAC, but not by LeuRS (Figure 5b) suggests that phosphoryl-transfer from ATP to the bound adenylate is obligatory (or, alternately occurs faster than) reversal of the activation reaction for the overall LeuAC catalytic cycle. The sign of the LeuAC-leucine interaction term (Figure 5c) implies that leucine stimulates PPi exchange by LeuAC less effectively than by full length LeuRS. The absence of full-length CP1, CP2, and ABD domains likely alters the range of conformational motion in LeuAC, as it does in the TrpRS urzyme [TrpAC; [45,46]].

These factors, together with the ~200-fold tighter binding of the aminoacyl group by LeuAC, relative to that observed for TrpAC could plausibly alter relative rates of PPi exchange and aa-5′AMP product release sufficiently that restoration of the transition-state configuration is the fastest route to turnover necessary for PP_i_ exchange, consistent with the increased steady-state rate of AMP production seen in Appendix A. Furthermore, if the bound Leu-5′AMP resides in a sub-optimal LeuAC conformation that is restored by its in situ *phosphorylation* to regenerate the transition state configuration, that could explain why turnover increases after the first round of catalysis (Appendix A).

Further work will be necessary to clarify these mechanistic details. Among the clues that must be followed up are (i) that AMP is generated ~20 times faster by LeuRS than by LeuAC in single turnover experiments (Table 3), (ii) a third phase with increased apparent turnover appears in the timecourse for AMP production by LeuAC (Appendix A), (iii) high ADP concentrations inhibit LeuAC active-site titration timecourses, and LeuAC itself exhibits no phosphatase activity on [^32^P] pyrophosphate (Appendix A).

**Figure 9 ijms-23-04229-f009:**
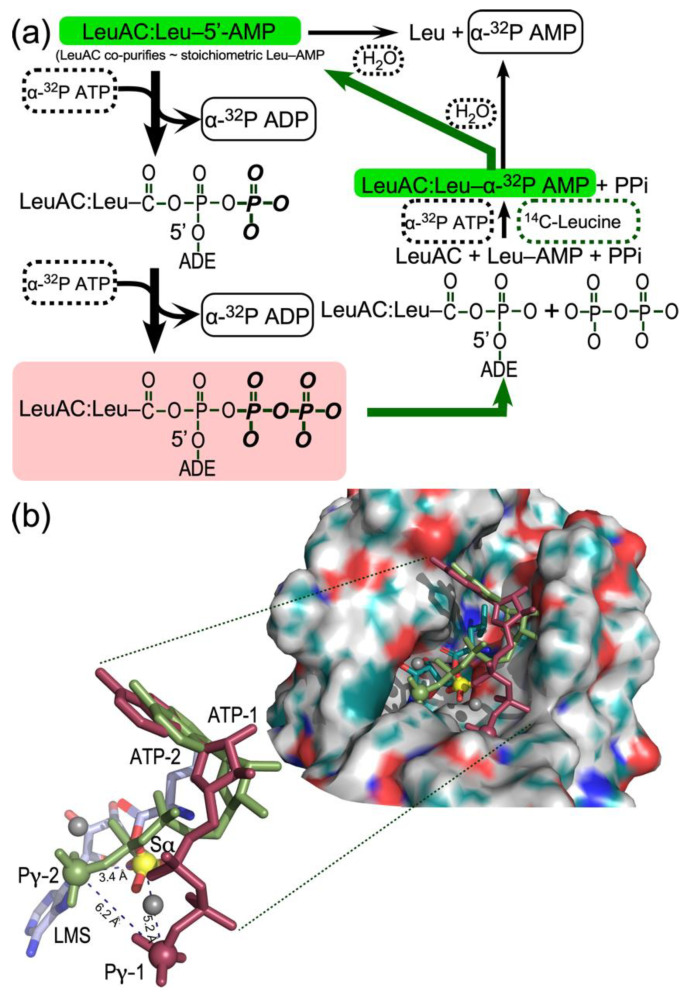
(**a**). Kinetic scheme accounting for formation of non-canonical sur-stoichiometric ADP (bold black arrows) and the incorporation of [^14^C] leucine into bound leucyl-5′AMP (bold green arrows, green background). Reactants—ATP, leucine, and water—are enclosed by dashed rectangles; products—ADP, AMP, and leucyl-5′AMP—by solid rectangles. ADE represents adenosine. The rose background highlights the fact that the second phosphorylation restores a configuration homologous to a dissociative transition state for leucine activation, and which thus would represent a mixture of substrates leucine and ATP and products Leu-5′AMP and PPi. (**b**). Evidence that ATP can bind in alternate configurations capable of phosphorylating both the α-phosphate (Sα) by Pγ-2 and the β-phosphate of the intermediate phosphorylated ADP by Pγ-1. The surface representation shows the LeuAC active site pocket (derived by energy minimization of LeuAC coordinates originally output from Rosetta [47], in which the bound leucyl-5′AMP is represented by leucyl-5′ sulfamoyl AMP (LMS). In addition, ATP molecules in which the glycoside and ribose-5′-phosphate torsion angles were reconfigured to the two most prominent configurations (ATP-1, red and ATP-2, green) observed in MD simulations of ATP in water without Mg^+2^ ion, as described in [43] and superimposed onto the adenine of the 3′-terminal adenosine of the crystallographic coordinates of tRNA^Leu^. The inset shows the relative positions in greater detail of the α-phosphate of LMS (yellow) and the putative γ-phosphates (Pγ-1; red) and (Pγ-2; green) of the two ATP molecules. These three atoms, represented as spheres, form a scalene triangle whose dimensions are consistent with the successive phosphorylation steps. Gray spheres are Mg^2+^ ions fitted during energy optimization.

### 3.4. Urzyme Catalysis Arises Largely from Secondary-Structural Conformation

The aaRS urzymes described to date are all quite sophisticated catalysts, accelerating the uncatalyzed rate for amino acid activation by ~10^9^-fold. Elimination of the KMSKS signature lysine residues reduce that acceleration by a small fraction of an order of magnitude. A rather profound implication of that limited impact on catalytic proficiency is that a major proportion of the rate acceleration by aaRS urzymes may not require specific amino acid side chains. That surprising independence of sequence—also suggested by the substantial sequence changes in LeuAC_2 (Appendix A)—raises the intriguing possibility that much of the rate acceleration by aaRS urzymes derives from secondary-structural conformational effects. Conversely, it suggests that their sophisticated biological activities are consistent with a much larger volume in amino acid sequence space than is the case with mature contemporary enzymes. Embedded within that idea is the notion that the evolution of catalytic activity began with quasi-species consistent with a reduced amino acid alphabet, and proceeded concomitantly with the growth of the coding alphabet and the speciation of the aaRS [4]. Validating that conclusion would substantially alter perceptions of the origin and evolution of catalysis by peptides, as we suggested originally on the basis of the activity observed from the bidirectional aaRS protozyme gene [12,15].

Moreover, this work illustrates how to pursue experimental validation of that hypothesis. Class I aaRS contain two signature sequences, HIGH and KMSKS, that furnish nearly all the recognizable catalytic interactions with ATP. The AMSAS mutant we have characterized here thus represents a second corner of the two-by-two thermodynamic cycle eliminating either KMSKS, HIGH, or both. Experimental characterization of the two remaining mutant LeuAC variants along lines outlined in Table 3 will allow measurement of the catalytic contributions of both signatures, together with their synergistic interaction, furnishing a quantitative estimate of the rate accelerations at all four corners, analogous to what we described earlier for the contributions of the CP1 and anticodon-binding domains to the catalysis by the TrpRS urzyme [46].

Finally, consistent with the notion that LeuAC catalysis arises predominantly from secondary-structural pocket formation, earlier work on the amino acid specificities of LeuAC and HisRS urzyme (see Figure 3 of [2]) suggest that LeuRS shows a wide range (~8000-fold) in its specificity constants (k_cat_/K_M_) for 19 of the 20 amino acids. Curiously, on average, LeuAC activates Class I in preference to Class II amino acid substrates approximately 4 times out of 5 and conversely for HisRS urzyme. Owing to the unusual difficulties with the acylation assay, we have not yet been able to make a comparison of tRNA specificies.

### 3.5. Realizing the Potential Utility of aaRS Urzymes Will Require More Thoughtful Re-Design

The dominant factor limiting more detailed studies of aaRS urzymes, especially structural studies [48], is their limited solubility. Data reported here for the LeuAC urzyme, together with Appendix A, suggest that soluble, active urzymes must be engineered more carefully, perhaps using newer and more effective Rosetta algorithms [49].

## 4. Materials and Methods

### 4.1. Expression and Purification of LeuRS and LeuAC

The gene for *Pyrococcus horikoshii* (Ph) LeuRS was synthesized by Gene Universal and expressed from pET-11a in BL21-CodonPlus (DE3)-RIPL (Agilent). Cells were grown at 37 °C and induced with 300 μM IPTG for 4 h then harvested and stored overnight at −20 °C. The cell pellet was resuspended in 1× Ni-NTA buffer (20 mM Tris, pH 8.0, 300 mM NaCl, 10 mM imidazole, 5 mM β-ME) plus cOmplete protease inhibitor (Roche) and lysed by three 15K psi passes on an Avestin Emulsiflex. Cell debris was pelleted at 4 °C 30 min 20K rpm. The soluble fraction was heated at 80 °C for 30 min to denature native *Escherichia coli* proteins. The heated cell extract was then pelleted, and the soluble material was loaded on to an equilibrated Ni-NTA column. The column was washed with three volumes 1x Ni-NTA buffer, then protein was eluted in a stepwise fashion with imidazole concentrations of 40, 80, 100, 200, and 500 mM imidazole. The fractions containing the protein of interest were pooled and dialyzed overnight against 200 mM HEPES, pH 7.4, 450 mM NaCl, 100 mM KCl, 10 mM β-ME. The following day the dialyzed protein was concentrated and mix to 50% glycerol and stored at −20 °C.

LeuAC was expressed as an MBP fusion from pMAL-c2x in BL21Star(DE3) (Invitrogen). Cells were grown, induced, harvested, and lysed similarly to Ph LeuRS with the distinct difference of being resuspended in Optimal Buffer (20 mM Tris, pH 7.4, 1 mM EDTA, 5 mM β-ME, 17.5% Glycerol, 0.1% NP40, 33 mM (NH_4_)_2_SO_4_, 1.25% Glycine, 300 mM Guanidine Hydrochloride) plus cOmplete protease inhibitor (Roche). LeuAC crude extract was then pelleted at 4 °C 30 min 15K rpm to remove insoluble material. The extract was then diluted 1:4 with Optimal Buffer and loaded onto equilibrated Amylose FF resin (Cytiva). The resin was washed with five column volumes of buffer and the protein was eluted with 10 mM maltose in Optimal Buffer. Fractions containing protein were concentrated and mixed to 50% glycerol and stored at −20 °C. All protein concentrations were determined using the Pierce™ Detergent-Compatible Bradford Assay Kit (Thermo Scientific). Experimental assays were performed either with the intact MBP-LeuAC fusion protein or with samples cleaved by tobacco etch virus (TEV) protease, purified as described [50]. Purity and cleavage efficiency was determined by running samples on PROTEAN^®^ TGX (Bio-RAD) gels. Some experiments used a second LeuAC variant, from a more recent Rosetta design algorithm, in which we attempted to modify surface residues to increase solubility (Matt Cummins, unpublished). Amino acid sequences for both variants are given in the Appendix A and compared with the native Ph LeuRS sequence in Supplemental Appendix A. We could not detect any significant differences between results from the two different LeuAC fusion constructs.

### 4.2. Nitrocellulose Filter Binding of the LeuRS/LeuAC Leucyl-Adenylate Complex

1 μM LeuRS or LeuAC was mixed with 25 μM [^14^C] leucine in 50 mM HEPES, pH 7.5, 20 mM MgCl_2_, 50 μM ATP, and inorganic pyrophosphatase. A zero time point was collected prior to addition of the enzyme. Reactions were incubated at 37 °C and aliquots were removed at indicated time points and spotted onto prewashed nitrocellulose filters. The filters were washed with reaction buffer lacking ATP and leucine, and allowed to dry before scintillation counting in 5 mL of BetaMax on a Beckman Packard.

### 4.3. Size Exclusion Chromatography of the LeuAC Leucyl-Adenylate Complex

1–8 μM of LeuAC was incubated at 37 °C for 36 min in buffer containing 50 mM HEPES, pH 7.5, 20 mM MgCl_2_, 4 mM ATP, and 40 μM [^14^C] leucine or 5 mM leucine containing comparable cpm of [^14^C] leucine. 2 ml of Sephadex G15 resin (Cytiva) was loaded into a disposable gravity flow column (120 × 10 mm) and equilibrated with five column volumes of 50 mM HEPES, pH 7.5 and 20 mM MgCl_2_. 100 μL of the reaction mix was applied to the column and eluted with 3 mL of buffer. Single-drop fractions (~40 μL) were collected and analyzed either by UV-Vis (Nanodrop) or scintillation counting (Beckman Packard). The obvious separation of LeuAC from lower molecular weight compounds enabled curve fitting to an exponentially modified normal distribution [30] with R^2^ values > 0.97 using JMP™ Pro. Quantitative estimation of the relative volumes of overlapping peaks was done using the eluted volume as the Y column and the corresponding value as the frequency in the JMP^TM^ Pro distribution module and fitting to a continuous 3-normal function.

### 4.4. Pyrophosphate (PPi) Exchange Assays

PPi exchange assays were performed as described [16,51], with some variation when we observed inhibition by KF. Enzyme concentrations were generally 3 μM (MBP LeuAC) and 0.3 μM (*E. coli* LeuRS) in 50 μL assay volumes. [^32^P] ATP formed at time intervals was separated from [^32^P] PPi by thin layer chromatography on polyethyleneimine plates (Scientific Adsorbents, Atlanta, GA, USA) that had been pre-run in water and developed using a mobile phase of 750 mm KH_2_PO_4_, pH 3.5 and 4 M urea, at a running temperature of 25 °C. Radioactive spots were detected and quantified by phosphorimaging.

### 4.5. Single Turnover Active-Site Titration Assays

Active-site titration assays were performed as described [16,51]. 3 μM of protein was added to 1× reaction mix (50 mM HEPES, pH 7.5, 10 mM MgCl_2_, 10 μM ATP, 50 mM amino acid, 1 mM DTT, inorganic pyrophosphatase, and either α- or γ-labeled [^32^P] ATP) to start the reaction. Timepoints were quenched in 0.4 M sodium acetate 0.1% SDS and kept on ice until all points had been collected. Quenched samples were spotted on (PEI) TLC plates, developed in 850 mM Tris, pH 8.0, dried and then exposed for varying amounts of time to a phosphor image screen and visualized with a Typhoon Scanner (Cytiva). The ImageJ measure function was used to quantitate intensities of each nucleotide. The time-dependence of loss (ATP) or de novo appearance (ADP, AMP) of the three adenine nucleotide phosphates were fitted using the nonlinear regression module of JMP™ Pro to Equation (1):Product(calc) = A × exp(−k_chem_ × seconds) − k_cat_ × seconds + C(1)
where k_chem_ is the first-order rate constant, k_cat_ is the rate of turnover, A is the amplitude of the first-order process, and C is an offset.

### 4.6. tRNA^Leu^ Aminoacylation Assays

A plasmid encoding the *P. horikoshii* tRNA^Leu^ (TAG codon) was synthesized by Integrated DNA Technologies and used as template for PCR amplification of the tRNA and upstream T7 promoter and downstream Hepatitis Delta Virus (HDV) ribozyme. The PCR product was used directly as template for T7 transcription. Following a 4 h transcription at 37 °C, the RNA was cycled five times (90 °C for 1 min, 60 °C for 2 min, 25 °C for 2 min) to increase the cleavage by HDV. The tRNA was purified by urea PAGE and crush and soak extraction. The tRNA 2′-3′ cyclic phosphate was removed by treatment with T4 PNK (New England Biolabs) following the manufacturer’s protocol. The tRNA was then phenol chloroform isoamyl alcohol extracted, filter concentrated, aliquoted and stored at −20 °C.

Aminoacylations were performed in 50 mM HEPES, pH 7.5, 10 mM MgCl_2_, 20 mM KCl, 5 mM DTT with indicated amounts of ATP and amino acids. Desired amounts of unlabeled tRNA—mixed with [α^32^P] A76-labeled tRNA for assays by LeuAC—were heated in 30 mM HEPES, pH 7.5, 30 mM KCl to 90 °C for 2 min. The tRNA was then cooled linearly (drop 1 °C/30 s) until it reached 80 °C when MgCl_2_ was added to a final concentration of 10 mM. The tRNA continued to cool linearly until it reached 20 °C.

Aminoacylation by *P. horikoshii* LeuRS followed incorporation of [^14^C] leucine into acid precipitable material. Enzyme was added to 0.95 μM and aliquots were taken at eight-minute timepoints and spotted onto 10% TCA prewashed Whatman 3MM filters, washed three times with 5 mL cold 5% TCA and a final wash of cold ethanol. Dried filters were counted in 5 mL BetaMax (MP BioMedicals) on a Beckman Packard scintillation counter.

[^14^C] leucine proved too insensitive for accurately measuring acylation by the weaker LeuAC catalyst, which was assayed by transfer of leucine to ‘3-terminal [^32^P] tRNA^Leu^. Re-folded tRNA was mixed with buffer and a zero timepoint collected prior to initiation of the reaction by addition of the enzyme to 1.2 μM. Eight-minute timepoints were quenched by adding into a solution of 0.4 M sodium acetate, pH 5.2, 6.25 mM Zn Acetate, 10U P1 nuclease and stored on ice until all timepoints had been collected. Quenched samples were incubated at 37 °C for 10 min to allow digestion of the tRNA by the P1 nuclease. Samples were spotted on pre-run PEI TLC plates and developed in 10% NH_4_Cl, 5% acetic acid.

Dried TLC plates were exposed overnight to a phosphor screen and visualized on a Typhoon Scanner. Integrated densities for AMP and leucyl-AMP were processed for analysis as single turnover experiments by computing first the fraction represented by leu-AMP. That fraction was multiplied by the total concentration of acylatable tRNA^Leu^ estimated from extended time courses with full-length *P. horikoshii* LeuRS to estimate the concentration of acylated product produced at each time point. Acylated product formation was then converted to the fraction of unacylated active substrate tRNA^Leu^ = (Total tRNA^Leu^ − acylated product)/Total tRNA^Leu^, and fitted to Equation (1) for single-turnover analysis.

Michaelis–Menten experiments were analyzed using either the standard Michaelis Menten equation or the improved version advocated by Johnson [52]. Johnson’s method fits the rectangular hyperbola using k_cat_ and the specificity constant k_SP_ = k_cat_/K_M_:v = kSP[S]/(1 + kSP[S]/kcat)(2)

The two nonlinear fitting equations gave almost exactly the same kinetic parameters, although Equation (2) gave better signal-to-noise estimates with noisy data.

### 4.7. Data Processing and Statistical Analysis

Phosphorimaging screens exposed to TLC plates were densitometered using ImageJ. Data were transferred to JMP™ Pro 16 via Microsoft Excel (version 16.49), after intermediate calculations. The nonlinear fitting module was used to fit all active-site titration and Michaelis–Menten datasets. Factorial design matrices in Table 1 and Table 2 were processed using the Fit model multiple regression analysis module of JMP™ Pro, using an appropriate form of Equation (3):Y_obs_ = β_0_ + Σβ_i_ × P_i_ + Σβ_ij_ × P_i_ × P_j_ + ε(3)
where Y_obs_ is a dependent variable, usually an experimental observation, β_0_ is a constant derived from the average value of Y_obs_, β_i_ and β_ij_ are coefficients to be fitted, P_i,j_ are independent variables used as predictors, and ε is a residual to be minimized. All rates were converted to free energies of activation, ΔG^‡^ = −RTln(k), before regression analysis because free energies are additive, whereas rates are multiplicative. For example, the activation free energy for the first-order decay rate in single-turnover experiments is ΔG^‡^k_chem_. It is worth emphasizing that multiple regression analyses of factorial designs exploit the replication inherent in the full collection of experiments to estimate the experimental variance on the basis of *t*-test P-values, in contrast to the presenting error bars showing the variance of individual datapoints. In some cases, multiple regression analyses reported here also entail experimental replicates, which enhance the associated analysis of variance.

### 4.8. Molecular Modeling

Torsion angle adjustments for different published ATP conformations were performed with UCSF Chimera [44], developed by the Resource for Biocomputing, Visualization, and Informatics at the University of California, San Francisco, with support from NIH P41-GM103311. Molecular superpositions were done using the CCP4 suite program LSQKAB [53] and visualized using PYMOL [54]. The energies of the resulting virtual complexes were optimized using the Sigma program [55]. The CHARMM force field was used to model the protein and the nucleic acids [56]. The generalized Born model [57] was used to describe the electrostatic interaction. 

## 5. Conclusions

The notion that far simpler aaRS drove early stages in the evolution of translation [7,8,9,10] enhanced understanding of how the Class distinction results from amino acid physical chemistry [58,59,60], how the operational code in the tRNA acceptor stem dictates aaRS recognition [18,19], and how aaRS secondary structural duality dictates amino acid side chain size discrimination [17]. Those valuable byproducts are independent of the authenticity of the reported catalytic activities of the TrpRS and HisRS urzymes and protozymes [2,3]. Nonetheless, as urzyme catalytic activities underpin the validation of the Rodin-Ohno hypothesis of bidirectional ancestral genetic coding [13,17], it is essential to validate them by creating new urzymes from other aaRS.

We characterize here a LeuRS urzyme from the subclass of Class I aaRS that require the most sophisticated editing machinery to discriminate between amino acids—leucine, isoleucine, and valine—that have most nearly the same size, and almost exactly the same hydrophobicity and thus are most difficult to distinguish. Work described here not only extends the repertoire of aaRS uryzmes by 50%; it also uncovers a potentially widespread catalytic promiscuity evidenced by the production of ADP, which appears to result from in situ phosphorylation of a tightly bound aminoacyl-5′AMP.

Notably, LeuAC’s enzymatic activities are relatively insensitive to mutation of catalytic residues in the active site, suggesting that largely sequence-independent conformational aspects were primarily responsible for the origins of catalysis. That important observation also implies that detailed positioning of specific catalytically important amino acid side chains followed later. These observations furnish a novel validation of the hypothesis of tight coupling between aaRS evolution and the development of the genetic coding table discussed elsewhere in this volume [6].

Finally, we address the range of experimental difficulties confronting work with such constructs. These difficulties—unpredictable stability, low solubility, and difficulty separating urzymes from macromolecular tags necessary to solubilize them—pose significant barriers to structural and mechanistic studies necessary to better understand their conformation and structure-function relationships. Addressing them will likely entail more insightful use of computational protein design.

## Figures and Tables

**Figure 1 ijms-23-04229-f001:**
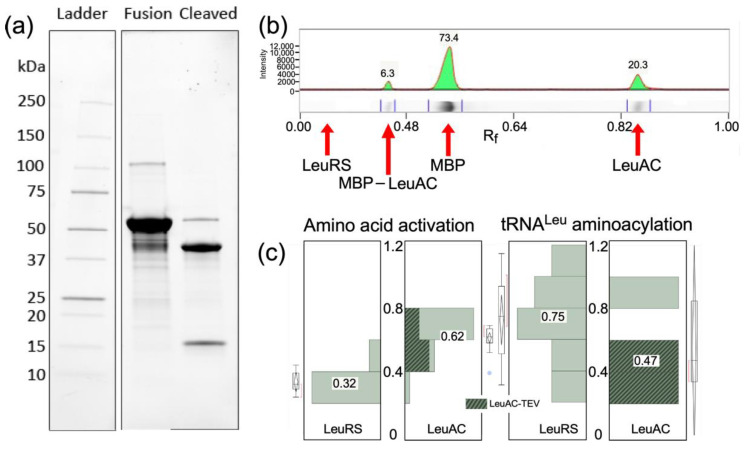
Purity of LeuAC urzyme. (**a**) PROTEAN^®^ TGX PAGE gel of purified LeuAC-MBP fusion protein and its TEV-cleaved products. Visualization is proportional to the tryptophan content of each band, as noted in the text. (**b**) Densitometric scan of the gel in (**a**). Integrated peak percentages are indicated for full-length LeuRS, LeuAC_MBP fusion, MBP, and LeuAC. (**c**) Distributions of amplitude parameters, A, estimated by fitting [^32^P] ATP consumption and tRNA^Leu^ aminoacylation time courses for LeuRS and TEV-cleaved LeuAC to Equation (1). Acylation experiments include the subset of experiments from Appendix A that most closely approximate single turnover conditions (5 > [substrate]/[enzyme] > 0.5). Median A values given as fractions against a white background are aligned with horizontal lines in the box plots. Median values for activation differ by 11.5 times the standard error of the mean for LeuRS (*n* = 13). For acylation the difference between median values for full length LeuRS and LeuAC, 0.27, is 3.7 times the standard error of the mean for LeuRS (*n* = 7). The difference for activation is both larger and inverted because LeuAC converts near stoichiometric amounts of ATP into ADP, in addition to AMP, as discussed in Section 2.3.2 and Section 3.3.

**Figure 2 ijms-23-04229-f002:**
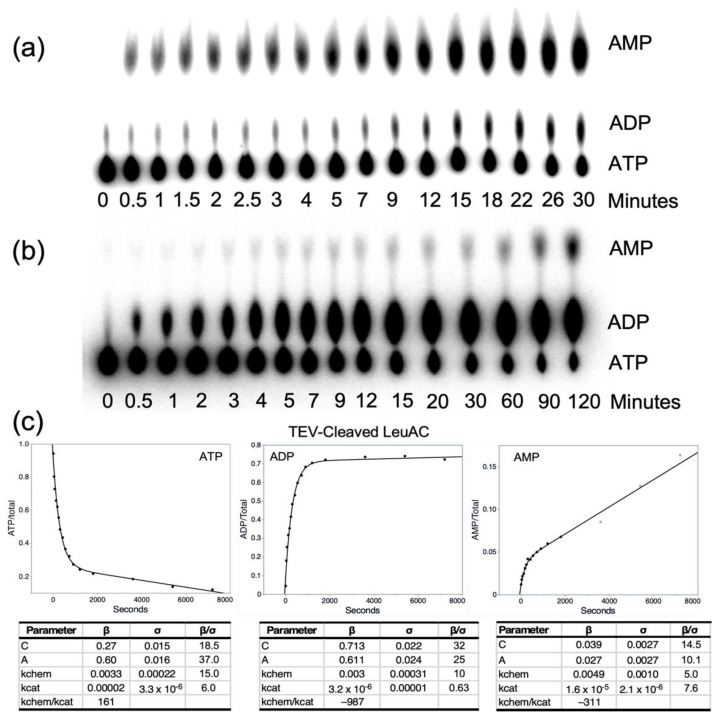
Time courses for the appearance of the three adenine nucleotides in active-site titration experiments with TEV-cleaved MBP-LeuAC fusion protein measured using [α-^32^P] ATP. Polyethyleneimine (PEI) thin-layer chromatograms (TLC) for full-length *P. horikoshii* LeuRS (**a**) and TEV-cleaved MBP LeuAC (**b**) were performed using [α-^32^P] ATP. Assays were performed as described in Section 4.5. Note the difference in timescales between the full length and urzyme proteins, as well as the different proportions of the appearance of [^32^P] ADP and [^32^P] AMP. (**c**) Plots of the timecourse in (**b**) fitted to Equation (1). Tables below each plot show the fitted parameters (β) their standard deviations (σ) and signal to noise.

**Figure 3 ijms-23-04229-f003:**
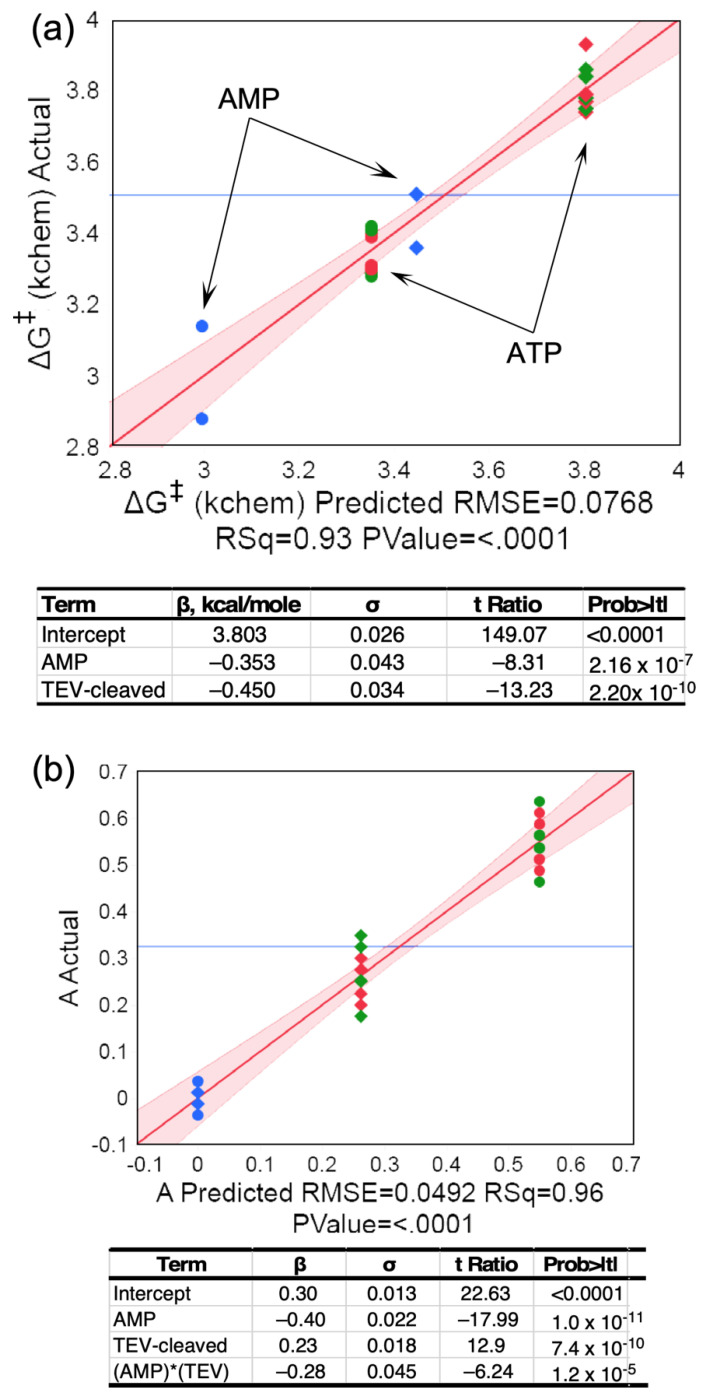
Multiple regression analyses of the dependence of ΔG^‡^k_chem_ and burst size on TEV cleavage and nucleotide. (**a**) Dependence of ΔG^‡^k_chem_ on nucleotide and TEC cleavage. The smaller activation free energy for AMP production by both catalysts (arrows) implies that reaction occurs faster. (**b**) Analysis of dependence of burst size ‘A’ on nucleotide and TEV cleavage. Data drawn from AST experiments in Table 1 and performed using both [α-^32^P] and [γ-^32^P] ATP. In both (**a**,**b**) circles represent TEV cleaved, diamonds fusion protein, blue and green parameters calculated from AMP and ADP production, respectively, and red parameters calculated from ATP consumption. R^2^ and overall F-ratio *p*-values are included in the × axis labels. Tables include β coefficients with σ values, Student *t*-tests, and *p* values.

**Figure 4 ijms-23-04229-f004:**
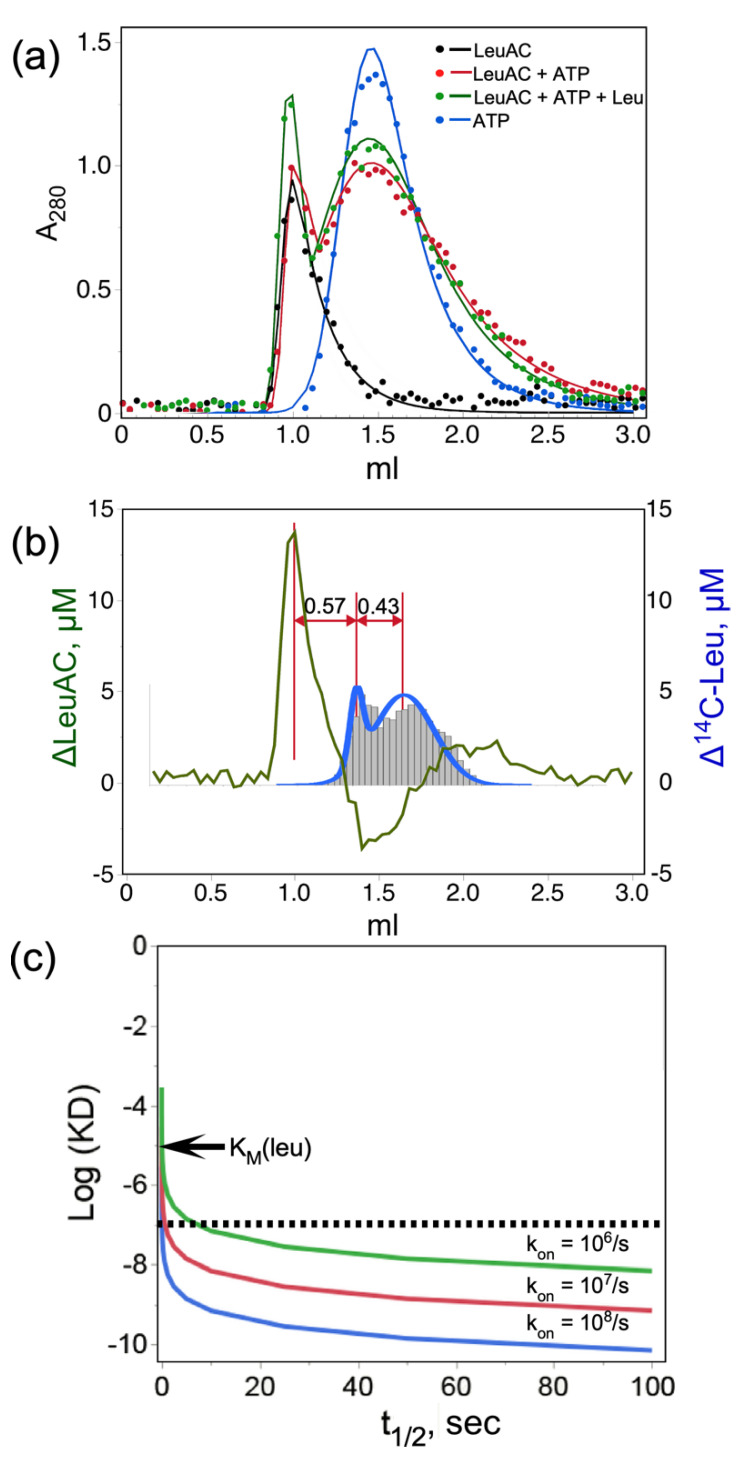
Quantitative size exclusion chromatography of TEV-cleaved LeuAC on G15 Sephadex. (**a**) Profiles for LeuAC, ATP, and reaction mixtures either with or without 5 mM leucine. Experimental points were fitted to a four-parameter exponentially modified normal distribution [30]. (**b**) Quantitative deconvolution of overlapping peaks in [LeuAC ± leucine (green)] and [leucine ± LeuAC (blue)] difference profiles show the mutual acceleration by LeuAC and substrate leucine of the other’s eluted position. (**c**) Simulated dissociation constants consistent with the displacement of bound [^14^C] leucine in (**b**). Values of log(K_D_), where K_D_ = k_off_/k_on_, are plotted against the half-life of the complex, t_1/2_ = 0.693 × k_off_, for reasonable values of k_on_. The observed displacement in (**b**) represents approximately 10 min or 40% of the total transit time in the G15 support. The abscissa covers an ~100-fold range ending at 100 s, or 1/6th of the actual displacement. Simulated curves are shown for three plausible on-rates. Essentially all possible values for K_D_ are less than 10^−7^ M, which is two orders of magnitude tighter than the K_M_ for leucine in Section 2.5.

**Figure 5 ijms-23-04229-f005:**
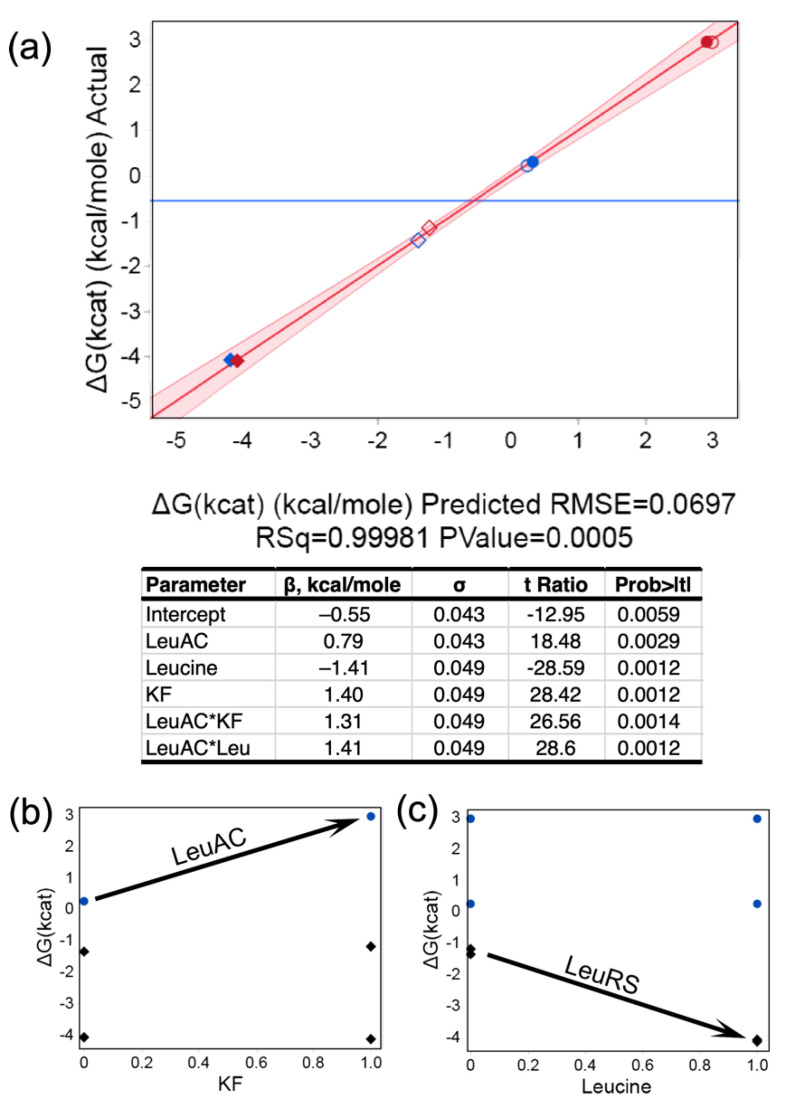
Regression analysis of PP_i_ exchange activities based on the design matrix in Table 2. (**a**) Multivariate model for the activation free energy, ΔG^‡^k_cat_, and table of β coefficients with standard deviations, Student’s *t*-tests, and *p*-value probabilities. Symbols: diamonds = LeuRS, circles = LeuAC, empty symbols = no added leucine, red symbols = plus KF. Supplemental plots of the effects of [leucine] (**b**) and KF (**c**) clarify the interpretation of two-way interactions, as noted in the text.

**Figure 6 ijms-23-04229-f006:**
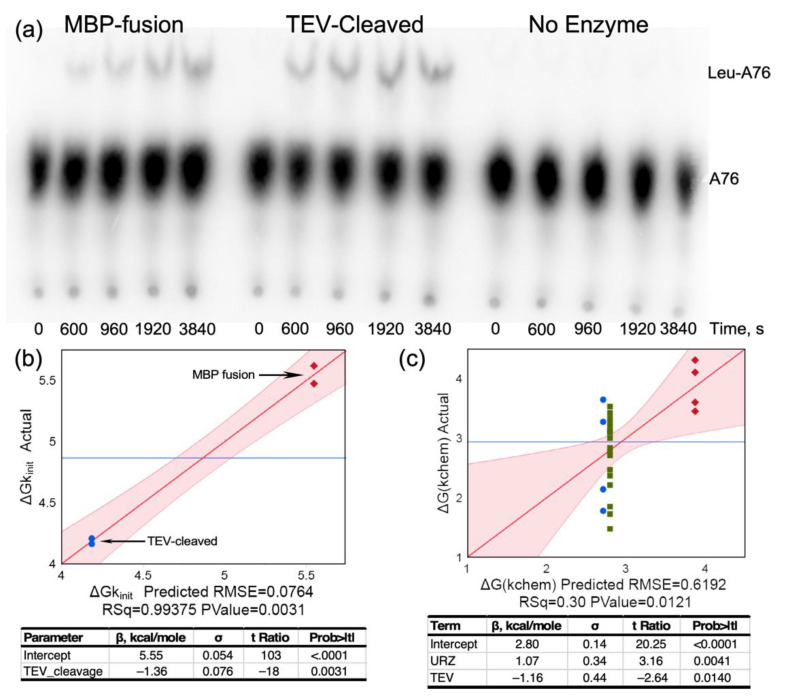
Aminoacylation of tRNA^Leu^ by LeuAC increases upon TEV cleavage of the MBP fusion. (**a**) TLC of time courses for the MBP fusion, TEV-cleaved LeuAC, and a minus enzyme control. (**b**) Comparison of the initial rates for matched assays similar to that in A on different days, with the corresponding regression analysis indicating a decrease of –1.36 in ΔG^‡^ for the initial rate, corresponding to an ~10-fold increase in rate. (**c**) A similar regression analysis of the dependence of the first-order rate, ΔG^‡^k_chem_, obtained from biphasic kinetic parameters for fitting time courses for the 28 acylation experiments in Appendix A. In (**b**,**c**), red diamonds are MBP fusion, blue circles are TEV-cleaved LeuAC, and in (**c**) green squares are full-length LeuRS.

**Figure 7 ijms-23-04229-f007:**
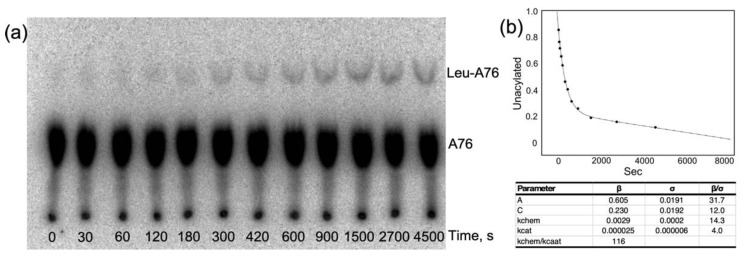
Analysis of a more finely divided time course for LeuAC aminoacylation, showing and excellent fit to a bi-phasic time course. The LeuAC used here is the MBP fusion. (**a**) TLC separation of acylated leucyl-2′,3′AMP as a function of time. (**b**) Time course in A fitted to Equation (1).

**Figure 8 ijms-23-04229-f008:**
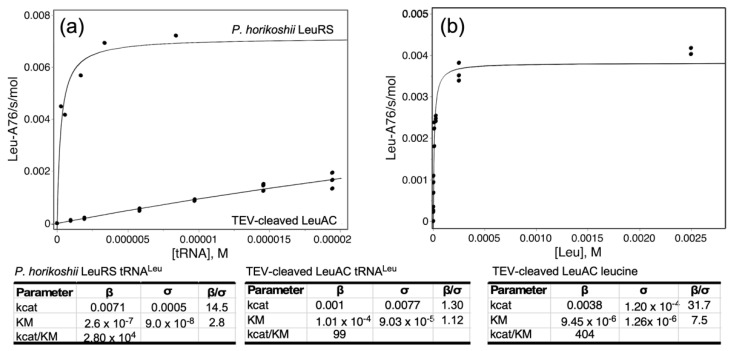
Steady-state kinetic analyses of *P.horikoshii* LeuRS and TEV-Cleaved LeuAC. (**a**) tRNA^Leu^-dependence. Results for LeuRS and LeuAC are plotted on the same graph to emphasize the qualitative difference in K_M_. (**b**) Leucine dependence. Fitted kinetic parameters are given in the tables, together with statistical significance. Measurements for LeuAC were done in triplicate.

**Table 1 ijms-23-04229-t001:** Design matrix for dependencies of [^32^P] ATP active site titrations. Independent variables are [ATP], [ADP], [AMP], TEV cleavage. Dependent variables—k_chem_, k_cat_, A, C, and associated free energies—are from fitting to Equation (1). LeuAC1 and LeuAC2 are labelled according to Appendix A.

Sample	ATP	ADP	AMP	TEV	k_chem_	ΔG^‡^ (k_chem_)	k_cat_	ΔG^‡^ (_kcat_)	C	A
MBP_LeuAC2	1	0	0	0	0.0017	3.79	2.7 × 10^−5^	6.24	0.6	0.2
MBP_LeuAC2, LSA	1	0	0	0	0.0017	3.77	2.2 × 10^−5^	6.35	0.49	0.3
MBP_LeuAC2, APQ	1	0	0	0	0.0018	3.74	1.7 × 10^−5^	6.52	0.48	0.3
MBP_LeuAC2	0	1	0	0	0.0015	3.86	2.3 × 10^−5^	6.33	0.37	0.2
MBP_LeuAC2, Vanadate	0	1	0	0	0.0015	3.84	1.2 × 10^−5^	6.73	0.5	0.3
MBP_LeuAC2, LSA	0	1	0	0	0.0018	3.75	2.0 × 10^−5^	6.41	0.48	0.3
MBP_LeuAC2, APQ	0	1	0	0	0.0017	3.78	1.4 x10^−5^	6.63	0.5	0.3
MBP_LeuAC2	0	0	1	0	0.0027	3.51	8.2 × 10^−7^	8.29	0.04	0
MBP_LeuAC2_Vanadate	0	0	1	0	0.0034	3.36	2.8 × 10^−6^	7.57	0.03	0
MBP_LeuAC1	1	0	0	0	0.0013	3.93	3.6 × 10^−6^	7.43	0.8	0.2
LeuAC1_Tev_cleaved	1	0	0	1	0.0033	3.39	0.00002	6.41	0.27	0.6
LeuAC1_Tev_cleaved	1	0	0	1	0.0032	3.40	2.1 × 10^−5^	6.38	0.26	0.6
LeuAC1_Tev_cleaved	0	1	0	1	0.0031	3.42	3.2 × 10^−6^	7.50	0.71	0.6
LeuAC1_Tev_cleaved	0	1	0	1	0.0032	3.41	2.1 × 10^−6^	7.74	0.69	0.6
LeuAC1_Tev_cleaved	0	0	1	1	0.0049	3.14	1.6 × 10^−5^	6.54	0.04	0
LeuAC1_Tev_cleaved	0	0	1	1	0.0077	2.88	1.9 × 10^−5^	6.43	0.03	0
LeuAC1_Tev-cleaved, Ile	1	0	0	1	0.0038	3.30	0.00003	6.16	0.36	0.5
LeuAC1_Tev-cleaved, no aa	1	0	0	1	0.0037	3.31	0.00003	6.16	0.34	0.5
LeuAC1_Tev-cleaved, Ile	0	1	0	1	0.0039	3.28	0.00001	7.06	0.6	0.5
LeuAC1_Tev-cleaved, no aa	0	1	0	1	0.0039	3.29	0.00001	6.88	0.62	0.5

**Table 2 ijms-23-04229-t002:** Design matrix for dependences of PPi exchange. Independent variables are LeuAC (1) or LeuRS (−1), [Leucine] (1,0) potassium flouride (0,1), and enzyme concentration. Dependent variables are the rate of ATP formation and derived quantities: kcat = rate/[Enz], ΔG^‡^ (k_cat_) = –RT ln(k_cat_). The LeuAC2 construct was used as the MBP fusion in these experiments.

EXPT	LeuAC	Leucine	KF	[Enz], M	Δ[ATP]/Time	k_cat_ (/s)	ΔG^‡^ (_kcat_) (Kcal/Mole)
1	1	0	0	3.00 × 10^−6^	2.02 × 10^−6^	6.7 × 10^−1^	0.234
2	1	0	1	3.00 × 10^−6^	2.06 × 10^−8^	6.9 × 10^−3^	2.949
3	1	1	0	3.00 × 10^−6^	2.00 × 10^−6^	6.7 × 10^−1^	0.239
4	1	1	1	3.00 × 10^−6^	2.07 × 10^−4^	6.9 × 10^−3^	2.945
5	−1	0	0	3.00 × 10^−6^	3.15 × 10^−6^	10.0	−1.391
6	−1	0	1	3.00 × 10^−6^	2.28 × 10^−6^	7.6	−1.201
7	−1	1	0	3.00 × 10^−6^	3.12 × 10^−4^	1.000	−4.112
8	−1	1	1	3.00 × 10^−6^	3.15 × 10^-4^	1.100	−4.119

**Table 3 ijms-23-04229-t003:** Summary data for comparing catalytic activities of WT LeuAC and the triple lysine mutant. Amino acid activation was assayed by active-site titration. tRNA^Leu^ aminoacylation was assayed with 1.6 μM LeuAC and 3 μM active tRNA^Leu^. Tables show regression coefficients, β. In kcal/mole for the dependence of the respective activation free energies (ΔG^‡^k_chem_ for activation; ΔG^‡^_obs rate_ for acylation) on changes in the respective independent variables, together with their *t*-test *p*-values.

Term	β. Kcal/Mole	σ	t Ratio	Prob>|t|
*Activation*				
Intercept	2.60	0.10	25	<0.0001
A	4.05	0.31	13	0.0002
KMSKS	−0.50	0.03	−18	0.00006
TEV	−0.56	0.04	−13	0.0002
(KMSKS) × (TEV)	−1.72	0.09	−19	0.00005
*Acylation*				
Intercept	5.11	0.05	103.7	<0.0001
KMSKS	−0.29	0.06	−4.7	0.003

**Table 4 ijms-23-04229-t004:** Variation of first-order and turnover rates for amino acid activation and tRNA^Leu^ aminoacylation by LeuRS, MBP-LeuRS and TEV-cleaved LeuRS. Mean values of ΔG^‡^k_chem/_k_cat_ = –RT ln(k_chem/_k_cat_) with standard deviations and numbers of assays are given for the different constructs.

Activation	All LeuRS	Ph LeuRS	Ec LeuRS	All LeuAC	MBP_fusion	TEV-Cleaved
<ΔG^‡^k_chem/_k_cat_>	−3.37	−3.81	−3.20	−2.45	−2.26	−2.60
Stdev	0.39	0.31	0.26	0.42	0.42	0.39
n	7	2	5	13	6	7
Aminoacylation						
<ΔG^‡^k_chem_k_cat_>		−2.81		−2.27	−1.97	−2.57
Stdev		0.71		0.77	0.69	0.82
n		20		8	4	4

## Data Availability

Data are provided in Appendix A.

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
