# Peer review of "A Leucyl-tRNA Synthetase Urzyme: Authenticity of tRNA Synthetase Catalytic Activities and Promiscuous Phosphorylation of Leucyl-5′AMP"

_ijms, 2022, doi:10.3390/ijms23084229_

Round 1

Reviewer 1 Report

Hobson et al followed up on previous studies on urzymes from the same group that centers around the hypothesis that ancient aminoacyl-tRNA synthetases (aaRS) were foundational enzymes during the origin of life. In these original enzymes, Class I and Class II aaRSs were encoded by the same DNA but read in different directions to provide distinct enzymes with catalytic activity. The authors fill a gap by presenting design and careful biochemical evaluation of a LeuRS variant of these urzymes based on LeuRS in one direction.

Conceptually, the idea of urzymes is extremely interesting, as it allows glimpses into a potentially ancient world. The work presented here combines protein engineering, careful biochemical characterization, and insightful evaluation to make the case that these urzymes are indeed feasible. Comments provided here can be addressed through discussion and no additional experiments are needed, if the data is not available already.

  1. I am curious about the specificity of LeuAC - does it discriminate between tRNAs if a bulk mixture if offered?
  2. On the same topic, LeuAC displays similar Kcats if offered isoleucine as with leucine if I am reading Table 1 correctly - how is activity towards amino acids that are chemically more distinct from leucine?

Minor points:

Line 346: the sentence sounds odd - might need reworking.

Author Response

We are grateful for reviewers generous reception of this manuscript and for their suggested improvements. We have, in addition, included a key new experimental determination of the effects of mutating the KMSKSK signature to AMSASA, which reduces both amino acid activation and acylation significantly. Finally, we have rewritten much of the manuscript to accommodate these new data.

Reviewer 1

Hobson et al followed up on previous studies on urzymes from the same group that centers

around the hypothesis that ancient aminoacyl-tRNA synthetases (aaRS) were foundational

enzymes during the origin of life. In these original enzymes, Class I and Class II aaRSs were

encoded by the same DNA but read in different directions to provide distinct enzymes with

catalytic activity. The authors fill a gap by presenting design and careful biochemical evaluation of

a LeuRS variant of these urzymes based on LeuRS in one direction.

Conceptually, the idea of urzymes is extremely interesting, as it allows glimpses into a potentially

ancient world. The work presented here combines protein engineering, careful biochemical

characterization, and insightful evaluation to make the case that these urzymes are indeed

feasible. Comments provided here can be addressed through discussion and no additional

experiments are needed, if the data is not available already.

  1. I am curious about the specificity of LeuAC - does it discriminate between tRNAs if a bulk

mixture is offered?

We have not measured LeuAC specificity for different tRNA substrates, which is nevertheless a key future goal now under investigation.

  1. On the same topic, LeuAC displays similar Kcats if offered isoleucine as with leucine if I am

reading Table 1 correctly - how is activity towards amino acids that are chemically more

distinct from leucine?

The reviewer correctly identifies that LeuAC reacts similarly with leucine and isoleucine. In fact, as shown in Figure 3 of Reference [2], a preliminary comparison shows a wide range (~8000-fold) in its proficiencies for 19 of the 20 amino acids. Curiously, on average, LeuAC activates Class I in preference to Class II amino acid substrates approximately 4 times out of 5. We have discussed this point in Section 3.4 of the revised Discussion.

Minor points:

Line 346: the sentence sounds odd - might need reworking.

Agreed; we have attempted clarification in the revision.

Reviewer 2 Report

There is very little one needs to say about this manuscript: it is excellent, comprehensive, carefully explained and in-depth throughout.. The amount of detailed experimental work is impressive. A landmark paper!

I have only a couple of minor suggestions for possible clarifications/improvements:

  1. It may be a good idea to describe the suggested mechanism regarding ADP production in the Abstract (line 25), even if the details still unclear. It is great to see the previous observations from the Zamecnik lab "dusted off" and brought to forefront once again.
  2.  In the Abstract, the authors talk about ATP configurations (line 27). Are they sure that they do not mean ATP conformations (in the sense that they are not fixed by covalent bonds in alternate structures)?
  3. Line 38: type: "Our previous work built [a/upon a?] new experimental framework ....
  4. I wonder whether showing a schematic figure of the various substructures as a schematic diagram (covering written descriptions in lines 99-105) would be helpful for extra clarity?
  5.  Figure 2 legend: For extra clarity, mentioning that the TLC plates are PEI plates could be useful (line 245/246)
  6. Line 281: typo: "reproducible"

Author Response

Reviewer 2

There is very little one needs to say about this manuscript: it is excellent, comprehensive,

carefully explained and in-depth throughout.. The amount of detailed experimental work is

impressive. A landmark paper!

I have only a couple of minor suggestions for possible clarifications/improvements:

  1. It may be a good idea to describe the suggested mechanism regarding ADP production in the

Abstract (line 25), even if the details still unclear. It is great to see the previous observations

from the Zamecnik lab "dusted off" and brought to forefront once again.

The revised abstract now clarifies the relationship between the ATP consumption and the ADP production.

  1. In the Abstract, the authors talk about ATP configurations (line 27). Are they sure that they do

not mean ATP conformations (in the sense that they are not fixed by covalent bonds in

alternate structures)?

We are grateful to the reviewer for catching this incorrect usage, which is corrected in the revision.

  1. Line 38: type: "Our previous work built [a/upon a?] new experimental framework ....

Another good catch, corrected in the revision.

  1. I wonder whether showing a schematic figure of the various substructures as a schematic

diagram (covering written descriptions in lines 99-105) would be helpful for extra clarity?

We have rewritten that paragraph for clarity and referenced a new graphical abstract, which is the suggested schematic. This was a very helpful suggestion!

  1. Figure 2 legend: For extra clarity, mentioning that the TLC plates are PEI plates could be

useful (line 245/246)

Agreed. We have modified the Figure 2 caption accordingly. Plates used for TLC are identified with manufacturer in the Methods section (Section 4.4).

  1. Line 281: typo: "reproducible"

Corrected.